# Analysis of Cadmium-Stress-Induced microRNAs and Their Targets Reveals bra-miR172b-3p as a Potential Cd$^{2+}$-Specific Resistance Factor in *Brassica juncea*

**Lili Liu** [1,2,*], **Hanqi Yin** [3], **Yanhui Liu** [1], **Lunhao Shen** [1], **Xiaojun Yang** [1], **Dawei Zhang** [1,2], **Mei Li** [4] **and Mingli Yan** [1,2,4]

[1] School of Life Science, Hunan University of Science and Technology, Taoyuan Road, Xiangtan 411201, China; liuyanhui2021@126.com (Y.L.); shenlunhao@163.com (L.S.); yangxiaojun6@126.com (X.Y.); zhangdawei@hnust.edu.cn (D.Z.); ymljack@126.com (M.Y.)
[2] Hunan Key Laboratory of Economic Crops Genetic Improvement and Integrated Utilization, Hunan University of Science and Technology, Taoyuan Road, Xiangtan 411201, China
[3] South China Institute of Biomedine, Guangzhou 510700, China; Hanqi.yin@longseemed.com
[4] Crop Research Institute, Hunan Academy of Agricultural Sciences, Changsha 410125, China; limei1230@126.com
[*] Correspondence: lilyliu@hnust.edu.cn; Tel.: +86-731-58291416

**Abstract:** The contamination of soil with high levels of cadmium (Cd) is of increasing concern, as Cd is a heavy metal element that seriously limits crop productivity and quality, thus affecting human health. (1) Background: Some miRNAs play key regulatory roles in response to Cd stress, but few have been explored in the highly Cd-enriched coefficient oilseed crop, *Brassica juncea*. (2) Methods: The genome-wide identification and characterization of miRNAs and their targets in leaves and roots of *Brassica juncea* exposed to Cd stress was undertaken using strand specific transcript sequencing and miRNA sequencing. (3) Results: In total, 11 known and novel miRNAs, as well as 56 target transcripts, were identified as Cd-responsive miRNAs and transcripts. Additionally, four corresponding target transcripts of six miRNAs, including *FLA9* (Fasciclin-Like Arabinogalactan-protein 9), *ATCAT3* (catalase 3), *DOX1* (dioxygenases) and *ATCCS* (copper chaperone for superoxide dismutase), were found to be involved in the plant's biotic stress pathway. We further validated the expression of three miRNA and six target genes in response to Cd, hydrargyrum (Hg), manganese (Mn), plumbum (Pb) or natrium (Na) stress and Mucor infection by qRT-PCR, and show that *ATCCS* and *FLA9* were significantly and differentially regulated in the Cd-treated leaves. In addition, our results showed that *DOX1* was obviously induced by Pb stress. Among the respective target miRNAs, bra-miR172b-3p (target for *ATCCS*) and ra-miR398-3p (target for *FLA9*) were down-regulated in Cd-treated leaves. (4) Conclusions: We identified bra-miR172b-3p as a potential Cd-specific resistant inhibitor, which may be negatively regulated in *ATCCS* in response to Cd stress. These findings could provide further insight into the regulatory networks of Cd-responsive miRNA in *Brassica juncea*.

**Keywords:** cadmium stress; miRNA; transcriptome; high-throughput sequencing; *Brassica juncea*

## 1. Introduction

Heavy metals have been found to be a kind of widespread pollutant and to pose a threat to organisms. Vast areas of agricultural soil are contaminated by heavy metals via atmospheric deposition or anthropogenic activities, such as animal manure, the direct application of phosphate fertilizers, sewage sludge, and irrigation water [1,2]. Cadmium (Cd), is one of the most prevalent and poisonous elements in agricultural soil, and jeopardizes the environment and human health. Toxicology studies have been undertaken of Cd stress in plants affecting tissue morphological, biochemical and physiological changes, as well as altered gene expressions [3–6]. The presence of 40 mg L$^{-1}$ Cd in soil has already been

found to affect yield in some crops [7]. Cadmium ions are easily taken in by plant roots and then transported to other organs, given Cd's high mobility and water solubility, and this severely disturbs physiological processes in plants [8] and inhibits plant development by affecting respiration, photosynthesis and nitrogen metabolism [9].

*Brassica juncea L.* (AABB, 2*n* = 36) is an amphidiploid species that originated from interspecies crosses between *Brassica nigra* (AA, 2*n* = 16) and *Brassica rapa* (BB, 2*n* = 20), which is another important annual or biennial oil crop and leafy vegetable used worldwide [10,11]. *Brassica juncea* is also a cruciferous plant capable of accumulating significant quantities of metals, including Cd, Pb, zinc (Zn), chromium (Cr), copper (Cu), aurum (Au) and selenium (Se) [12–14]. The plant can accumulate >400 µg Cd/g dry weight in its leaves over 96 h, and it therefore holds promise for the remediation of Cd-contaminated soils, given its high biomass, moderate metal accumulation capacity, ease of harvesting, and metal tolerance [15].

MicroRNAs (miRNAs) are a class of small endogenous non-coding regulatory RNA molecules containing approximately 18–30 nucleotides (nt), which can regulate gene expression through guiding translational repression and can also target mRNA cleavage, mainly at the post-transcriptional level [16–18]. An increasing number of studies have indicated that miRNAs play important regulatory in the response to heavy metals (Cd iron) in different species, such as *Brassica napus* [5,19], *Arabidopsis* [20,21], *Ipomoea aquatic Forsk* [22], *Typha angustifolia* [23], *Raphanus sativus L.* [24], maize [25], wheat [26,27] and rice [28–30]. Exploring Cd-regulated gene expression and the regulatory relationship between miRNA and target genes are both areas of interest as regards the molecular mechanism of metal homeostasis and accumulation [21,27,31]. Numerous studies have indicated the importance of miRNAs in biotic and abiotic stress responses to ion metals at the post-transcriptional level. For example, 39 differentially expressed miRNAs were identified via a comprehensive analysis of miRNA expression profiles after Cd treatment in *Brassica napus* [19]. Moreover, a Cd treatment of *Brassica napus* significantly affected the expression of 22 miRNAs belonging to 11 families in the roots, and 29 miRNAs belonging to 14 miRNA families in the shoots, of *Brassica napus* [5]. In total, 39 known and eight novel miRNAs were aberrantly expressed in response to Cd treatment in rice [28]. About 199 miRNAs were identified in xylem sap from maize seedlings, including 97 newly discovered miRNAs and 102 known miRNAs [25]. Plants overexpressing miR156 accumulated significantly less Cd in the shoots than the roots, and showed higher tolerance against Cd-stress [20]. The miR398 is involved in oxidative stress tolerance via the regulation of its target CSD, which is active in wheat seedlings exposed to Cd stress [26]. miR166 and miR268 act as negative regulators, modulating Cd tolerance and accumulation in rice [29,30]. miRNA395 is an up-regulator that enhances Cd retention and detoxification in the roots of *Ipomoea aquatic Forsk*, and miR5139, miR1511 and miR8155 contribute to Cd's translocation into the shoots of low-shoot-Cd cultivars [22]. This evidence strongly suggests the key roles of miRNAs in plants' Cd stress responses, and comprehensive analyses of plants' transcriptome and miRNA profiles are widely used to screen important miRNAs involved in biotic stress.

Our previous study has investigated the physiological responses of *Brassica juncea* L. to 30 and 50 mg kg$^{-1}$ Cd stress, and demonstrated the activities of catalase enzymes, the contents of soluble sugar and chlorophyll were reduced, the content of the soluble protein malondialdehyde increased [6]. Moreover, comparative transcriptomic analysis indicated that the downregulation of *HMA3* and the upregulation of *Nramp3*, *HMA2* and *Nramp1* also play roles in reducing Cd toxicity in the roots of *Brassica juncea* L. under Cd stress. However, the molecular activity of Cd in response to miRNAs and their corresponding pathways in *Brassica juncea* have yet to be fully elucidated. A combined analysis of microRNA and mRNA expression to infer Cd-induced regulation has not been performed for *Brassica juncea*. In the current study, with the aim of identifying the Cd-regulated unigenes and miRNAs in roots and leaves and developing a Cd-associated miRNA regulatory network, we constructed mRNA and sRNA libraries of Cd-treated and Cd-free *Brassica juncea*, which were then sequenced via an Illumina Hiseq2500 system. These results could provide a new

perspective for studying the regulation of miRNAs in root and leaf under Cd stress, and facilitate genetic breeding of Cd tolerance in *Brassica juncea*.

## 2. Materials and Methods

### 2.1. Plant Culture and Cd Treatment

Seeds of *Brassica juncea* (collected from the Biological Park, Hunan University of Science and Technology, Xiangtan, China) were surface-sterilized with 75% ethanol for 1 min and mercuric chloride for 15 min, followed by rinsing and soaking in sterile distilled water. After germination at 22 °C for 3 days, the seeds were transferred to a growth chamber with modified half-strength (MS) solid medium for 16 h at 22 °C (light), 8 h at 16 °C (dark). The concentration of Cd in dry weight of crop leaves was more than 5–10 $\mu$g Cd g$^{-1}$, which was toxic to plants organs [32]. To examine the impact of Cd contamination on the expression patterns of mRNAs and miRNAs, plants grown in MS solid medium were supplemented with 0, 15, 30, 45 or 60 mg L$^{-1}$ Cd$^{2+}$ for 30 days, and seedlings grown in Cd$^{2+}$-free MS solid medium were used as the control.

Previous studies have demonstrated that a high concentration of Cd (40 mg L$^{-1}$; 50 mg Kg$^{-1}$) causes damage to seedlings by changing their cellular metabolic activity [6,7]. In this study, seedlings treated with 30 mg/L Cd were subjected to RNA sequencing or RT-PCR analysis. The roots and shoots were separately harvested and immediately frozen in liquid nitrogen. Meanwhile, seeds grown in MS medium containing NaCl (Na$^+$, 3.45 g L$^{-1}$), PbNO$_3$ (Pb$^+$, 800 mg L$^{-1}$), HgCl (Hg$^+$, 30 mg L$^{-1}$) and MnSO$_4$ (Mn$^{2+}$, 200 mg L$^{-1}$), and Mucor of positive infection (0.1 mL of spore suspension, obtained by the dilution of *Mucorracemosus* to $10^2$ cfu mL$^{-1}$, was added in the culture medium) were used in the qRT-PCR analysis. Leaves were harvested after 30 d. There were six biological replicates in each experiment, with three technical repetitions.

### 2.2. High-Throughput Sequencing of Transcriptome and Small RNAs

Total RNA was isolated from each pooled sample using Trizol reagent (Invitrogen, Carlsbad, CA, USA) according to the manufacturer's protocol. The RNA's quantity and quality were determined via an Agilent 2100 Bioanalyzer (Agilent, MN, USA). The transcriptome library was prepared using an Illumina TruSeq Stranded Total RNA LT Sample Prep Kit (Illumina, CA, USA) following the manufacturer's protocols. In short, 3 $\mu$g purified total RNA was used for extracting coding RNA and non-coding RNA using a Ribo-Zero kit, which can remove ribosomal RNA. Then, the isolated RNA was eluted and fragmented in Elute-Prime-Fragment Mix at 94 °C for 8 min, followed by first-strand and second-strand cDNA syntheses, and end-repair and dA-tailing ligation. To construct four small RNA (sRNA) libraries from the roots and leaves, sRNAs of 18–30 bp were separated and gel-purified from approximately 5 ug total RNAs each group. Then, 5′ and 3′-Illumina RNA adapters were ligated to the isolated sRNAs using T4 RNA ligase, and the adapter-ligated sRNAs were reverse-transcribed and amplified for 15 cycles with PCR. Both the transcriptome and sRNAs were sequenced on the Illumina HiSeq2500.

### 2.3. Transcriptome Assembly and Unigene Annotation

Raw RNA-seq reads were processed by the ShortRead package to remove some PCR primer sequences, adapters, and the low-quality nucleotides [33]. Reads less more than 2 N and longer than 35 nt (ambiguous nucleotides) were retained. Moreover, the paired reads that mapped to the SILVA database (http://www.arb-silva.de/download/arb-files/, accessed on January 2020) were discarded. The clean reads of each sample were then assembled as unigenes using the Trinity package, with –SS_lib_type FR and an optimized k-mer length of 25 [34,35]. The assembled unigenes of all samples were cleaned by removing redundancies, and further assembled as all-unigenes using CD-HIT software [36]. The cleaned reads of each sample were aligned to the assembled all-unigenes using FANSe2 [37], allowing for 7 nt mismatches [38]. The all-unigenes with at least 10 mapped reads were considered reliably assembled unigenes.

The Cluster of Orthologous Groups (COG) and Kyoto Encyclopedia of Genes and Genomes database (KEGG) pathway annotations were performed via a BLASTX search against UniProt (UniProtKB/TrEMBL Invertebrate Protein Database), the KEGG, and the COG database with an E-value of $1e^{-5}$. Gene ontology (GO) analysis, which is a form of functional analysis that associates differentially expressed genes with GO categories, was performed on our data. In each annotated sequence, successful blast hits were strand-specifically mapped and annotated using Blast2GO for all the unigenes. The GO analysis of the all-unigenes was performed using the WEGO software [39,40].

### 2.4. Identification of Differentially Expressed Unigenes

The read numbers of each unigene were first transformed into Reads Per Kilo bases per Million reads (RPKM), and then differentially expressed unigenes in Cd-treated plants were identified by the DEGseq package using an MA-plot-based method with Fisher's exact test (FET) [33]. $p < 0.05$ and |fold change| > 2 were used as the threshold to judge the significance of the unigene expression difference between the treatment and the control. To identify significantly enriched GO and pathway terms, all unigenes showing significant differences in terms of transcript abundance between the two groups were mapped to the GO databases with Blast2GO software and plant pathway databases with MapMan software [24].

### 2.5. Bioinformatic Analysis of Small RNA Sequencing Data

Based on clean reads from two independent small RNA libraries (removed from impurity and low quality in the raw reads), the unique small RNAs, ranging from 18 to 30 bp, were mapped to the *Brassica juncea* reference sequences, which contain mRNA transcriptome sequences, genomic survey sequences (GSS) and expressed sequence tag (EST) sequences deposited in NCBI databases. These unique sRNA sequences were compared with RNA family (Rfam) databases with the BLASTn search program. Some non-coding RNAs were then filtered out, including ribosomal RNAs (rRNA), transfer RNAs (tRNAs), small nucleolar RNAs (snoRNAs), and small nuclear RNAs (snRNAs)

After identifying known miRNAs by mapping the clean reads to the Nt, Rfam and miRBase databases (http://www.mirbase.org/index.shtml, accessed on March 2020), the remaining unannotated sRNAs were used to screen the novel miRNAs in the MIREAP software (https://sourceforge.net/projects/mireap/, accessed on March 2020), according to the following parameters: the allowed miRNA sequence length ranged from 18 to 30 bp, and the allowed reference sequence length ranged from 20 to 23 bp. The previously reported basic criteria were also employed in the novel miRNA prediction [41]. According to the instructions for determining high-confidence miRNAs in miRBase 21, the predicted novel miRNAs were further screened and validated in this study. The secondary structures of pre-miRNA for novel candidates were constructed by the Mfold software [42].

### 2.6. Differential Expression Analysis of Cd Treatment-Related miRNAs

The small RNA clean reads were then aligned to known miRNA sequences from other plant species deposited in miRBase 21. The expression abundance of all the identified miRNAs was normalized in each library (normalized expression = actual miRNA count/total count of clean reads × 1,000,000, transcripts per million (TPM)). The differential expression of $Cd^{2+}$ treatment-related miRNAs between the four libraries was calculated as follows: fold-change = log2 (miRNA normalized reads in "T-Leaf"/miRNA normalized reads in "C-leaf" or "T-Root"/miRNA normalized reads in "C-Root"). The *p*-value was calculated via previously established methods [43]. miRNAs with log2|fold-change| ≥ 1 and $p \leq 0.05$ were considered as up- or down-regulated miRNAs in response to Cd stress, respectively.

### 2.7. Prediction and Annotation of Target Genes for miRNAs

The potential target prediction of differentially expressed miRNAs was conducted with the plant small RNA target server or web-based resource [44]. Some analysis parameters

were set as the default for predicting potential miRNA target genes, as follows: (1) no more than four mismatches between identified miRNAs and target genes; (2) no mismatch between positions 10 and 11; (3) no more than two consecutive mismatches.

### 2.8. qRT-PCR Validation

Reverse transcription–quantitative real-time PCR (qRT-PCR) was performed to validate the quality of the high-throughput-sequencing and the relative quantitative expression levels of miRNAs and targets from roots and leaves cultivated with different metal ions ($Cd^{2+}$, $Hg^{2+}$, $Mn^{2+}$, $Pb^{2+}$, and $Na^+$) or Mucor infection, as previously described. The total RNAs from three biological replicates were isolated and treated with DNase I, and then transcribed to cDNA following the manufacturer's protocol [41]. The amplification reactions were carried out on a BioRad iQ5 sequence detection system (BioRad, USA), following the reported protocol [6]. The specific primers of target genes for qRT-PCR were designed using Beacon Designer 7.0 (Premier Bio-soft International, USA). All the primer sequences of PCRs are listed in Supplementary Table S1. The relative gene expression data were analyzed using the $2^{-\Delta\Delta CT}$ method. 5.8S rRNA and GAPDH were used as the reference genes.

## 3. Results

### 3.1. High-Throughput Sequencing of Transcriptome and sRNA Response to $Cd^{2+}$ Stress

To obtain a comprehensive overview of the *Brassica juncea* transcriptome, mRNA libraries constructed from the total RNAs of the leaves and roots were sequenced via the HiSeq2500 system, resulting in the generation of 48,486,320 and 42,168,040 raw reads for the leaves and roots, respectively (Table 1). After the removal of poly (A) tails, short and low-quality tags, and adaptor contamination, 47,383,313 and 41,228,691 clean reads were obtained, respectively. After further pair-end annotation and gap filling, 126,072 unigenes were assembled with an average length of 1770 bp and an N75 length of 975 bp (Table 2). The root and leaf transcriptome libraries combined with available GSS and EST sequences made up the reference sequences of *Brassica juncea* for the prediction of novel miRNAs.

**Table 1.** Statistical analysis of sequencing reads from RNA-seq and miRNA-seq libraries of *Brassica juncea L.* leaves and roots.

| Category | RNA-seq | | | | miRNA-seq | | | |
|---|---|---|---|---|---|---|---|---|
| | C-Leaf | T-Leaf | C-Root | T-Root | C-Leaf | T-Leaf | C-Root | T-Root |
| Raw Reads | 46,653,696 | 48,486,320 | 43,480,452 | 42,168,040 | 21,966,292 | 22,022,242 | 22,330,210 | 24,168,375 |
| Clean Reads | 45,524,282 | 47,383,313 | 42,495,034 | 41,228,691 | 16,143,364 | 16,240,568 | 17,621,183 | 18,108,357 |
| rRNA Trim | 42,887,556 | 39,793,164 | 40,859,852 | 39,793,164 | 16,140,557 | 16,237,736 | 17,618,404 | 18,105,841 |
| Clean Ratio | 91.9% | 82.1% | 94.0% | 94.4% | 73.5% | 73.7% | 78.9% | 74.9% |

**Table 2.** Assembly statistics for the *Brassica juncea L.* transcriptome.

| N25 | N50 | N75 | Longest | Mean | Median | Shortest | Unigenes | Uniprot Blast | ATH AGI Blast |
|---|---|---|---|---|---|---|---|---|---|
| 2233 | 1536 | 975 | 16,408 | 1770 | 1536 | 201 | 126,072 | 106,227 | 101,520 |

To screen the response of miRNAs to Cd in *Brassica juncea*, four sRNA libraries were constructed from the Cd-free and Cd-treated leaves and roots of *Brassica juncea* (T-root, C-root, T-leaf and C-leaf), and these were sequenced on the platform of a Hiseq 2500. Totals of 1,443,355, 1,115,351, 1,797,217 and 745,941 unique sequences were obtained from the C-leaf, T-leaf, C-root and T-root libraries, respectively (Table 3). In total, 6230 (C-leaf), 6749 (T-leaf), 10,716 (C-root) and 9974 (T-root) unique sequences were successfully mapped to reference sequences. After removing the non-coding rRNAs, tRNAs, snRNAs and snoRNAs, 245 (C-leaf), 245 (T-leaf), 254 (C-root) and 207 (T-root) uniquely matched known

miRNAs were obtained by querying the remaining sequences against miRbase. There were 1,437,125 (C-leaf), 1,108,602 (T-leaf), 1,786,501 (C-root) and 735,967 (T-root) unannotated unique sRNA sequences available for use to predict potentially novel miRNAs.

**Table 3.** The distribution of small RNAs among different categories in the leaves' and roots' libraries.

| Category | C-Leaf | | T-Leaf | | C-Root | | T-Root | |
|---|---|---|---|---|---|---|---|---|
| | Unique | Counts | Unique | Counts | Unique | Counts | Unique | Counts |
| Total sRNA | 1,443,355 | 10,581,384 | 1,115,351 | 9,229,340 | 1,797,217 | 14,053,530 | 745,941 | 13,365,075 |
| miRNA | 245 | 959,224 | 245 | 807,224 | 254 | 957,135 | 207 | 255,389 |
| rRNA | 4766 | 365,486 | 5541 | 347,275 | 9739 | 702,492 | 9299 | 932,729 |
| tRNA | 1185 | 416,521 | 944 | 231,685 | 639 | 87,037 | 434 | 32,504 |
| snRNA | 27 | 64 | 16 | 55 | 66 | 794 | 26 | 167 |
| snoRNA | 7 | 25 | 3 | 3 | 18 | 218 | 8 | 15 |
| Unannotated | 1,437,125 | 8,840,064 | 1,108,602 | 7,843,098 | 1,786,501 | 12,305,854 | 735,967 | 12,144,271 |

### 3.2. Functional Annotation of Unigenes

Annotation of the *Brassica juncea* transcriptome sequences was performed against two public databases, which were the Arabidopsis thaliana protein database and Uniprot protein database, using the BLASTX algorithm with a significant E-value threshold of $1 \times 10^{-5}$ (Figure 1a). Based on protein annotation and the *E*-value distribution, 25.5% and 29.8% of the mapped sequences showed strong homology (*E*-value $< 10^{-5}$) to available plant sequences in the Uniprot protein database and the Arabidopsis thaliana protein database, respectively, while 20.0% and 20.6% of mapped sequences showed very strong homology (*E*-value $< 10^{-50}$). Nearly 90% of unigenes can be annotated from five top-hit species, including *Brassica napus*, *Brassica rapa* subsp., *Brassica juncea*, *Eutrema salsugineum* and *Arabidopsos thaliana* (Figure 1b).

To understand the biological roles of unigenes, GO and pathway analyses were performed using Blast2GO. In each of the three "GO classification" categories, including biological process (BP), cellular component (CC) and molecular function (MF), which was mainly involved in metabolic process, cell and binding, respectively. Moreover, cellular process, cell part and catalytic were also well represented (Figure 2). However, few genes were assigned to the aspects such as locomotion, virion part and protein tag.

Functional classification and pathway assignment were performed via COG and KEGG. In total, 82,718 unigenes were classified into 25 COG categories (Figure 3), among which signal transduction mechanisms represented the largest group (10,587), followed by general function prediction, posttranslational modification, protein turnover and transcription. Meanwhile, coenzyme transport and metabolism, extracellular structures and cell motility were the smallest groups.

### 3.3. Identification of Known miRNAs in Brassica juncea

miRNAs are a class of small non-coding regulatory RNAs molecules with the length distribution of 18–30 nt [42,44]. In this study, the length of sRNAs ranged from 18 to 29 nt, and the majority of the total sequence reads were sRNAs of 21 nt in length (Figure 4). To screen the known miRNAs in the leaves and roots of *Brassica juncea*, the unique sequences were aligned to the mature sequences of known miRNAs deposited in miRBase. Among these miRNA families, miR156, miR169 and miR166 are the largest in the leaf and root. Besides this, miR9408 was only found in the leaf, while miR169_6 and miR857 were only found in the root (Figure 5a). The read number dramatically differed among the 43 miRNA families identified in the root and leaf libraries. miR166, miR156, miR159 and miR168 showed notably high expression levels in both libraries. miR166 was the most abundant gene family, with a normalized frequency in both libraries, followed by miR158 (Figure 5b).

Several miRNA families, such as miR398, miR403 and miR1885, were moderately abundant. Moreover, different miRNA families displayed significantly different expression levels in two libraries, such as miR169_6 and miR857, which were identified in only one library.

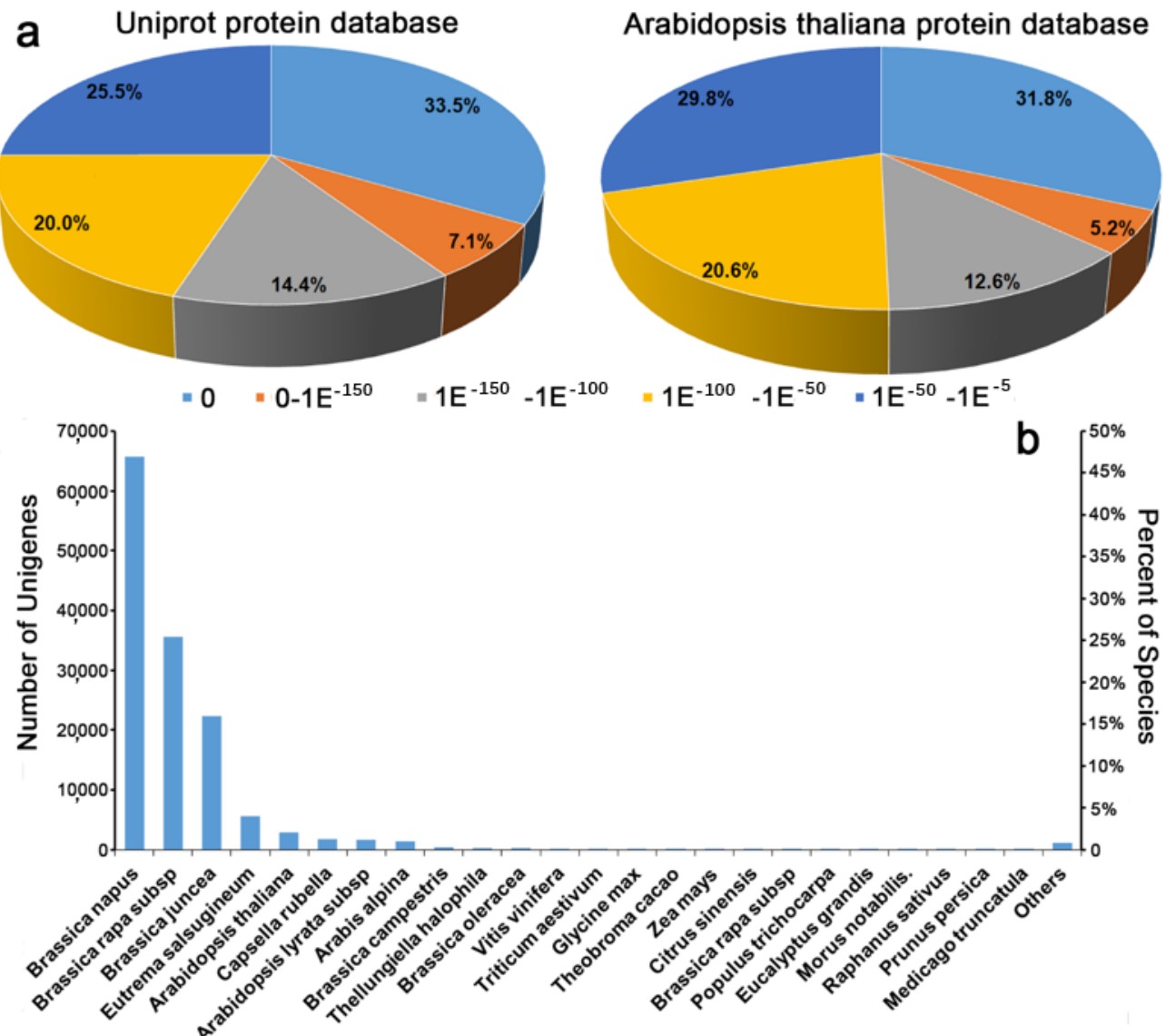

**Figure 1.** Blast *E*-value and distribution of top-hit species for transcriptome sequence: (**a**) Blast *E*-value on Uniprot protein database and Arabidopsis thaliana protein database; (**b**) the 25 top-hit species based on Uniprot annotation.

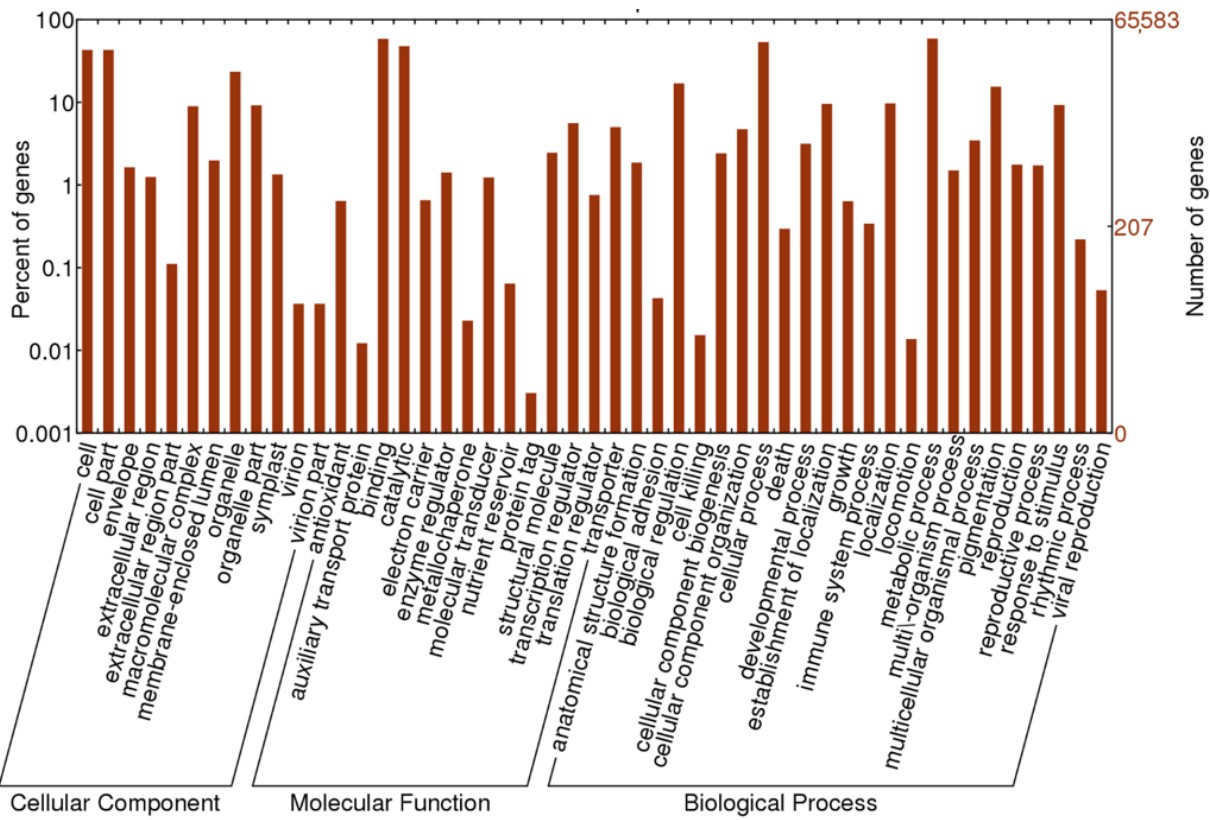

**Figure 2.** Gene Ontology classification of assembled unigenes in *Brassica juncea*. The results are summarized in the categories of cellular component, biological process and molecular function. The right *Y*-axis represents the number of genes and the left *Y*-axis shows the percentage of total unigenes.

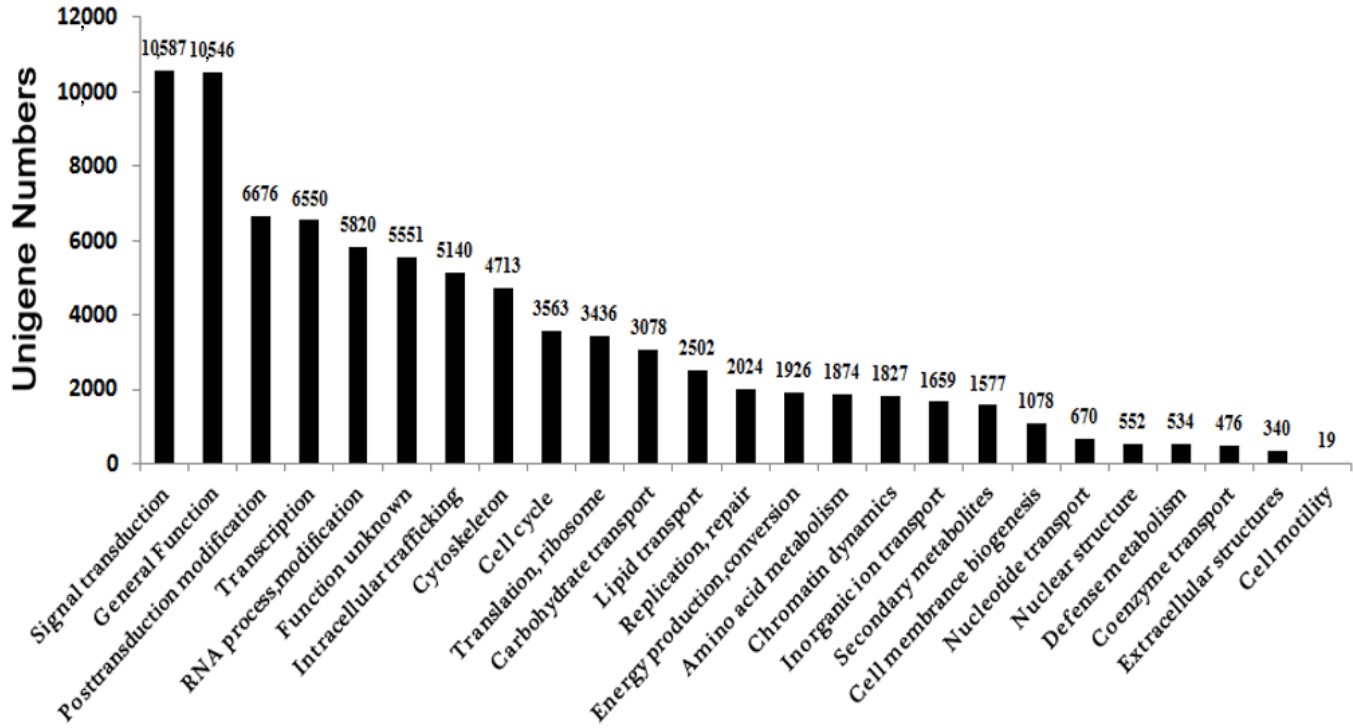

**Figure 3.** COG annotations of *Brassica juncea* unigenes. In total, 82,718 unigenes were classified into 25 COG categories.

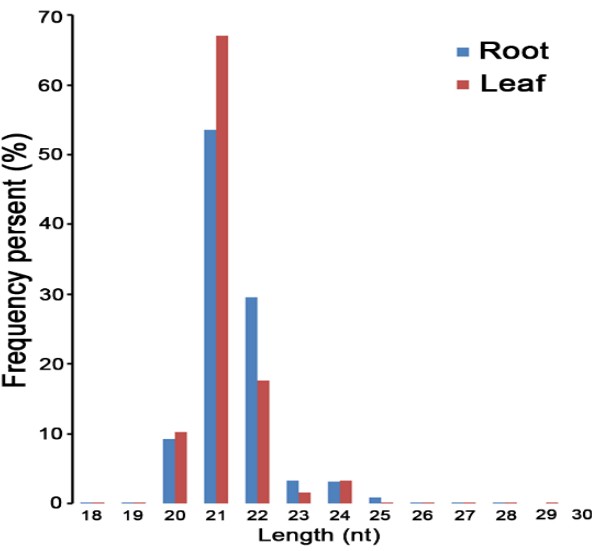

**Figure 4.** Length distribution of miRNAs identified from *Brassica juncea*.

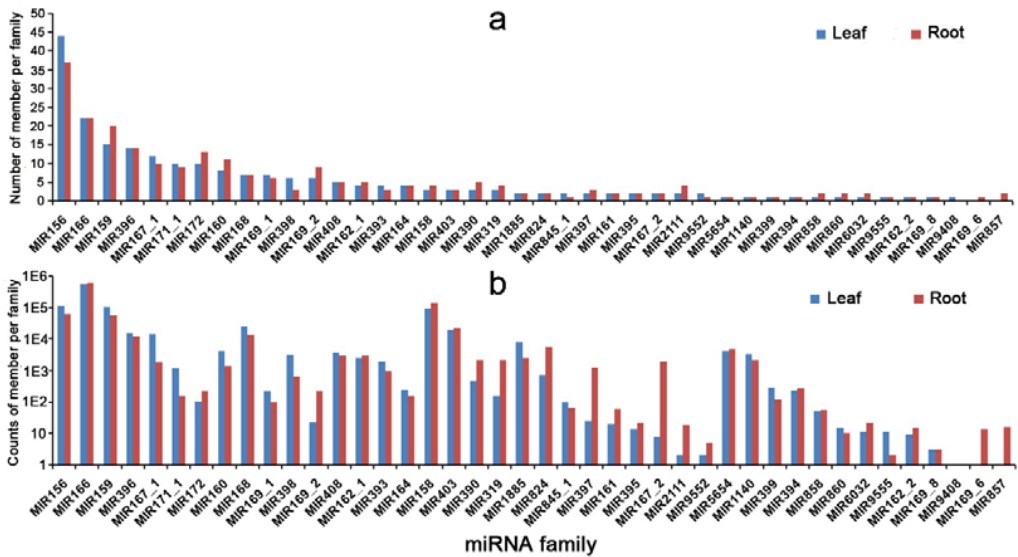

**Figure 5.** Sizes and abundance of identified known miRNA families from leaves and root of *Brassica juncea*: (**a**) Distribution of known miRNA family size in *Brassica juncea*; (**b**) Log2 (counts) of each known miRNA family from leaves and roots of *Brassica juncea*.

### 3.4. Identification of Cd-Responsive Unigenes and miRNAs in Brassica juncea

After calculation, the expression of each unigene was determined. We found that the number of down-regulated unigenes (1417) was larger than that of up-regulated unigenes (514) identified in T-leaf vs. C-leaf (Figure 6a,c). The numbers of up-regulated unigenes (312) and down-regulated unigenes (273) identified between T-root and C-root were similar. To explore the biological roles of unigenes, GO enrichment analysis was conducted (Figure 6b). The unigenes in leaves were mainly involved in photosynthesis and oxidation reduction, whereas the unigenes in roots were significantly related to the response to oxidative stress and the chitin catabolic process after Cd treatment, compared with controls.

To identify differentially expressed miRNAs under Cd stress in *Brassica juncea*, known and novel miRNAs were subjected to differential expression analysis between T-leaf and C-leaf, and between T-root and C-root, respectively. A total of 152 differentially expressed miR-NAs, including 59 known and 93 novel, were identified and considered as Cd-responsive miRNAs (Figure 6c).

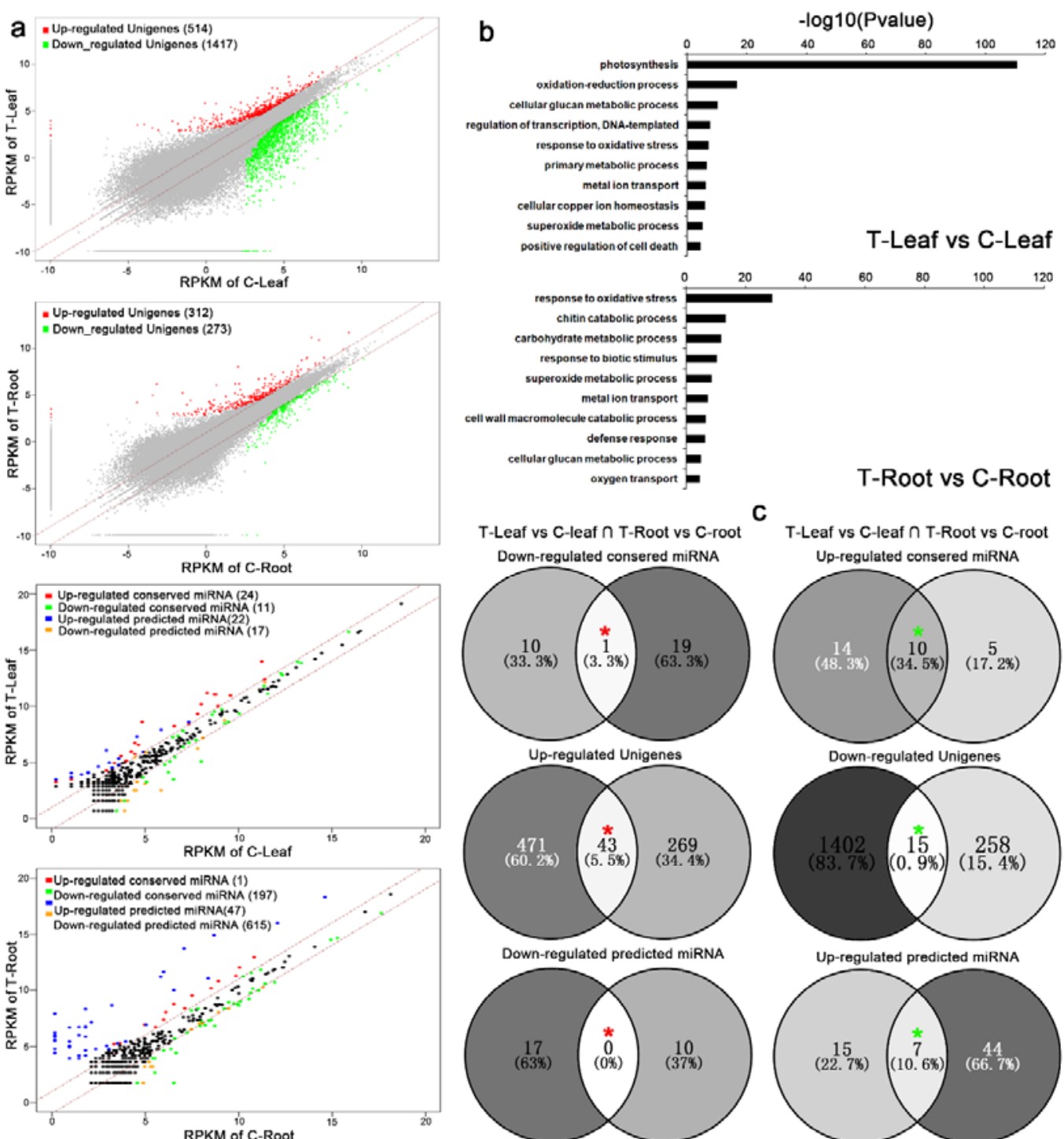

**Figure 6.** Identification and annotation of differentially expressed unigenes and miRNAs from leaves and roots with or without Cd treatment in *Brassica juncea*: (**a**) Changes of transcript abundance levels in leaves and roots with or without $Cd^{2+}$ treatment. (**b**) Functional annotation of differentially expressed unigenes in leaves and roots with or without Cd treatment. (**c**) The number of identified up- and down-regulated unigenes, and the known and novel miRNAs in leaves and roots of *Brassica juncea*, the red * sign means the numbers of intersection in leaf and root with down-regulated miRNA and negative(up)-regulated unigenes; the green * sign means the numbers of intersection in leaf and root with up-regulated miRNA and negative(down)-regulated unigenes.

### 3.5. Target Prediction of Cd-Responsive miRNAs

To functionally characterize the biological roles of miRNAs that respond to Cd stress, the psRNATarget server was applied to identify target genes for known and novel miR-NAs. A total of 27 target sequences for four conserved miRNAs (bra-miR408-5p, bra-miR397b, bra-miR398-3p and bra-miR172b-3p) were identified using MapMan, among which AT3G04320, AT1G03870 (fasciclin-like arabinogalactan protein 9, FLA9), AT1G20620 (catalase 3, *ATCAT3*), AT3G01420 (dioxygenases, *DOX1*), AT1G12520 (copper chaperone for superoxide dismutase, *ATCCS*) and AT1G73260 (*ATKT1*) were associated with biotic stresses on biological pathways (Table 4). Among these miRNAs, several were involved in translational inhibition, while some were related to target mRNA cleavage via regulating target sequences. Additionally, 29 target genes for seven novel miRNAs (mireap-3258, mireap-2590, mireap-3259, mireap-3479, mireap-2593, mireap-3553 and mireap-2592) were also predicted, among which AT1G03870 was related to biotic stresses on biological pathways (Tables 5 and 6).

**Table 4.** Identified target genes for conserved miRNAs in *Brassica juncea* L.

| miRNA | Target Sequence | Target Gene | Description | Inhibition |
|---|---|---|---|---|
| bra-miR408-5p | blank_rootc30873_g3_i1 | AT3G04320 | Kunitz family trypsin and protease inhibitor protein | Cleavage |
| | blank_leafc31640_g1_i4 | AT5G41080 | a member of the GDPD family | Translation |
| | blank_rootc32568_g1_i4 | AT3G47340 | a glutamine-dependent asparagine synthetase | Cleavage |
| | blank_leafc28232_g1_i1 | AT1G30720 | FAD-binding Berberine family protein | Translation |
| | blank_leafc31751_g3_i2 | AT1G21100 | O-methyltransferase family protein | Cleavage |
| bra-miR397b | blank_rootc33048_g2_i5 | AT3G16240 | Delta tonoplast intrinsic protein | Translation |
| | Experi_leafc29467_g1_i3 | AT1G80380 | a glycerate kinase | Cleavage |
| | blank_leafc32085_g1_i2 | AT3G47340 | a glutamine-dependent asparagine synthetase | Translation |
| bra-miR398-3p | Experi_leafc23645_g1_i1 | AT1G03870 | fasciclin-like arabinogalactan-protein 9 | Translation |
| | blank_rootc30463_g1_i4 | AT1G80380 | a glycerate kinase | Translation |
| | Experi_leafc30568_g1_i1 | AT1G61520 | PSI type III chlorophyll a/b-binding protein (Lhca3*1), cell-to-cell mobile | Translation |
| | Experi_rootc31467_g1_i3 | AT1G61520 | PSI type III chlorophyll a/b-binding protein (Lhca3*1), cell-to-cell mobile | Translation |
| bra-miR172b-3p | Experi_rootc35086_g1_i9 | AT3G60140 | a protein similar to beta-glucosidase | Cleavage |
| | Experi_leafc32122_g1_i2 | AT1G20620 | Catalase, catalyzes the breakdown of hydrogen peroxide ($H_2O_2$) | Translation |
| | Experi_leafc25017_g1_i2 | AT2G01890 | a purple acid phosphatase (PAP) | Cleavage |
| | Experi_leafc30056_g1_i1 | AT3G01420 | an alpha-dioxygenase | Translation |
| | Experi_leafc22299_g1_i2 | AT1G49470 | Family of unknown function (DUF716) | Translation |
| | Experi_leafc24092_g1_i1 | AT1G12520 | Copper-zinc SOD copper chaperone | Cleavage |
| | blank_rootc25533_g1_i1 | AT1G12520 | Copper-zinc SOD copper chaperone | Cleavage |
| | blank_leafc21081_g1_i1 | AT3G04720 | a protein similar to the antifungal chitin-binding protein | Translation |
| | Experi_leafc24092_g1_i4 | AT1G12520 | Copper-zinc SOD copper chaperone | Cleavage |
| | blank_rootc25533_g1_i2 | AT1G12520 | Copper-zinc SOD copper chaperone | Cleavage |
| | Experi_rootc22458_g2_i1 | AT1G73260 | a trypsin inhibitor, modulating programmed cell death | Translation |
| | blank_leafc30331_g2_i4 | AT5G39610 | a NAC-domain transcription factor | Translation |
| | Experi_leafc30462_g1_i2 | AT1G53990 | Contains lipase signature motif and GDSL domain | Translation |
| | blank_leafc16527_g1_i1 | AT4G04830 | methionine sulfoxide reductase B5 (MSRB5) | Translation |
| | Experi_rootc2154_g1_i2 | AT2G05520 | a glycine-rich protein in stems and leaves | Translation |

Genes related to the biotic stress pathway are underlined.

To further explore the roles of Cd-responsive miRNAs, the relevant biological pathways were conserved, and novel miRNAs that may play key roles in Cd response were depicted (Figure 7a). We found that six miRNAs were related to the abiotic stress pathway,

including bra-miR398-3p and mireap-2590 which target FLA9; mireap-2590, bra-miR408-5p and bra-miR398-5p, which target AT3G04320, and bra-miR172b-3p, which targets *DOX1*, *ATCCS*, *ATCAT3* and *ATKT1*.

**Table 5.** Novel miRNAs related to Cd resistance.

| miRNA ID | Length | Sequence | Position | TPM | | | |
|---|---|---|---|---|---|---|---|
| | | | | **C-Root** | **T-Root** | **C-Leaf** | **T-Leaf** |
| bra-mireap-3258 | 20 nt | GCGUGCUCAGGGCGUCGGCA | 3p | 78.1 | 193.5 | 64.7 | 215.3 |
| bra-mireap-2590 | 20 nt | GGCUAAGUCCGUUCGGUGGA | 5p | 113.6 | 281.7 | 8.2 | 59.9 |
| bra-mireap-3259 | 22 nt | GUGCUUGGCAGAAUCAGCGGGG | 5p | 4.0 | 9.8 | 4.1 | 15.9 |
| bra-mireap-3479 | 21 nt | GCAUGCUCAGGGCGUCGGCCU | 3p | 10.9 | 26.9 | 9.2 | 28.1 |
| bra-mireap-2593 | 21 nt | CCCAGUUCUGAACCCGUCGAC | 3p | 70.2 | 173.9 | 182.9 | 344.9 |
| bra-mireap-3553 | 20 nt | GCGUGCUCAGGGCGUCGGUC | 3p | 1.0 | 2.4 | 0.0 | 6.1 |
| bra-mireap-2592 | 23 nt | GGAUUGGCUCUGAGGGCUGGGCU | 5p | 4.0 | 9.8 | 6.2 | 22.0 |

**Table 6.** Identified target genes for novel miRNAs in *Brassica juncea* L.

| miRNA | Target Sequence | Target Gene | Description | Inhibition |
|---|---|---|---|---|
| mireap-3258 | Experi_leafc19595_g1_i1 | AT4G22520 | Bifunctional inhibitor/lipid-transfer/seed storage 2S albumin superfamily protein | Cleavage |
| | blank_leafc31640_g1_i4 | AT5G41080 | a member of the GDPD family | Cleavage |
| | blank_rootc32568_g1_i4 | AT3G47340 | a glutamine-dependent asparagine synthetase | Translation |
| | Experi_leafc29467_g1_i3 | AT1G80380 | a glycerate kinase | Cleavage |
| | blank_leafc32085_g1_i2 | AT3G47340 | a glutamine-dependent asparagine synthetase | Translation |
| mireap-2590 | blank_rootc30463_g1_i4 | AT1G80380 | a glycerate kinase | Translation |
| | Experi_leafc23645_g1_i1 | AT1G03870 | fasciclin-like arabinogalactan-protein 9 | Translation |
| | Experi_leafc29467_g1_i3 | AT1G80380 | a glycerate kinase | Cleavage |
| mireap-3259 | Experi_rootc31467_g1_i3 | AT1G61520 | PSI type III chlorophyll a/b-binding protein | Cleavage |
| | blank_leafc28232_g1_i1 | AT1G30720 | FAD-binding Berberine family protein | Translation |
| | blank_rootc32568_g1_i4 | AT3G47340 | a glutamine-dependent asparagine synthetase | Cleavage |
| mireap-3479 | blank_leafc31751_g3_i2 | AT1G21100 | O-methyltransferase family protein | Cleavage |
| | Experi_leafc29467_g1_i3 | AT1G80380 | a glycerate kinase | Translation |
| | blank_rootc32568_g1_i4 | AT3G47340 | a glutamine-dependent asparagine synthetase | Translation |
| | blank_leafc32085_g1_i2 | AT3G47340 | a glutamine-dependent asparagine synthetase | Translation |
| mireap-2593 | Experi_rootc31467_g1_i3 | AT1G61520 | PSI type III chlorophyll a/b-binding protein | Translation |
| | blank_leafc31640_g1_i4 | AT5G41080 | a member of the GDPD family | Translation |
| | blank_leafc31751_g3_i2 | AT1G21100 | O-methyltransferase family protein | Cleavage |
| mireap-3553 | Experi_leafc19595_g1_i1 | AT4G22520 | Bifunctional inhibitor/lipid-transfer protein/seed storage 2S albumin superfamily protein | Cleavage |
| | blank_leafc31640_g1_i4 | AT5G41080 | a member of the GDPD family | Cleavage |
| | blank_rootc32568_g1_i4 | AT3G47340 | a glutamine-dependent asparagine synthetase | Translation |
| | Experi_leafc29467_g1_i3 | AT1G80380 | a glycerate kinase | Cleavage |
| mireap-2592 | blank_leafc31751_g3_i2 | AT1G21100 | O-methyltransferase family protein | Cleavage |
| | blank_rootc32568_g1_i4 | AT3G47340 | a glutamine-dependent asparagine synthetase | Translation |
| | blank_rootc30463_g1_i4 | AT1G80380 | a glycerate kinase | Cleavage |
| | Experi_leafc29467_g1_i3 | AT1G80380 | a glycerate kinase | Cleavage |
| | blank_leafc32085_g1_i2 | AT3G47340 | a glutamine-dependent asparagine synthetase | Translation |
| | blank_leafc31640_g1_i4 | AT5G41080 | a member of the GDPD family | Cleavage |
| | blank_rootc30873_g3_i1 | AT3G04320 | Kunitz family trypsin and protease inhibitor protein | Cleavage |

Genes related to the biotic stress pathway are underlined.

### 3.6. Validation of Cd-Specific Resistant miRNA and Target Genes

To evaluate the validity of Illumina sequencing results, and further assess the regulation of resistant miRNAs and their target genes involved in Cd stress in *Brassica juncea*, the

expression levels of six selected target genes—*ATCCS*, AT3G04320, *FLA9, DOX1, ATCAT3* and *ATKT1*, and three miRNAs, including bra-miR172b-3p, mireap-2590 and bra-miR398-3p, in response to Cd, Hg, Mn, Pb and Na stress and Mucor infection were validated via qRT-PCR analysis, then compared with the normal control (Figure 7b). Among these, *ATCCS, FLA9* and *ATCAT3* showed more significant responses to heavy metal contamination. Moreover, *ATCCS* and *FLA9* were significantly and differentially up- or down-regulated in the Cd-treated leaves, respectively. *DOX1* was obviously induced by $Pb^{2+}$. Among the respective target miRNAs, bra-miR172b-3p (target for *ATCCS*) and bra-miR398-3p (target for *FLA9*) were down-regulated in Cd-treated leaves. However, bra-miR398-3p was oppositely regulated when assessed via the RNA-seq data. Based on the results of miRNA-seq and RNA-seq, bra-mir172b-3p was down-regulated, and its target genes *DOX1, ATCCS, ATCAT3* were significantly up-regulated expression in *Brassica juncea* under Cd stress, which was in line with the results by qRT-PCR analysis (Figure 7b,c). Therefore, we identified bra-miR172b-3p as a potential Cd-resistant inhibitor in *Brassica juncea*, the downregulation of which may release ATCCS in response to Cd stress.

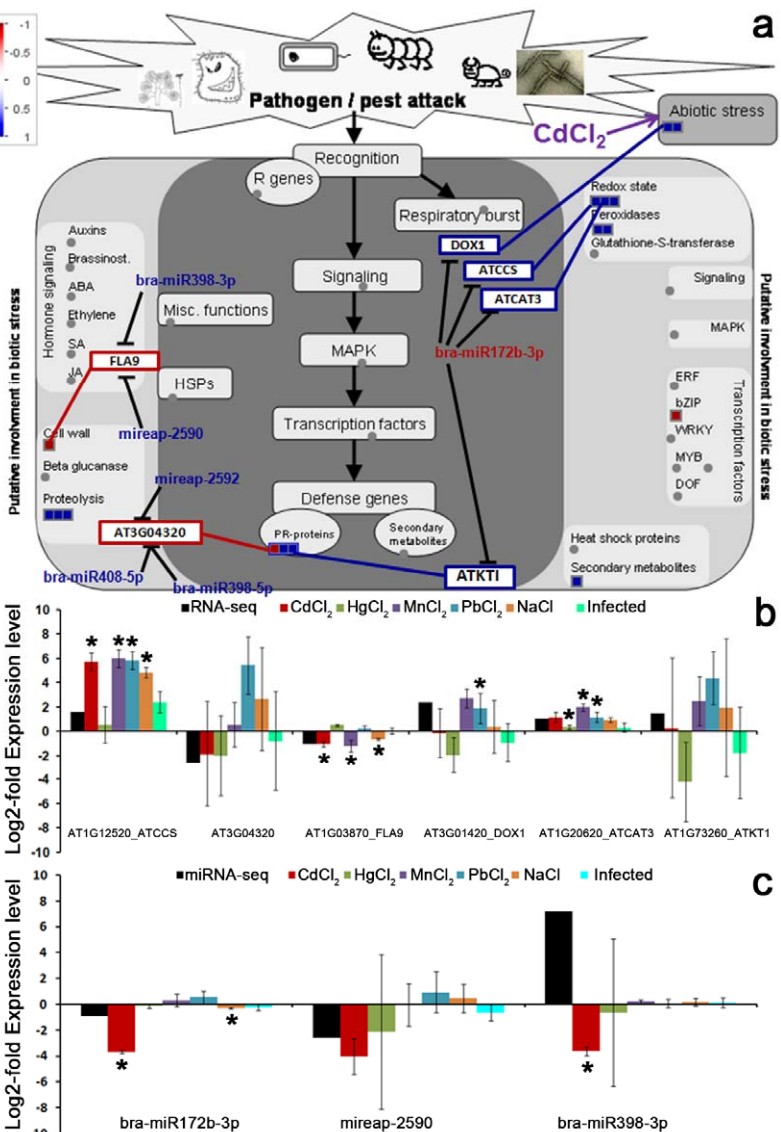

**Figure 7.** Verification of Cd stress in *Brassica juncea* and Cd-responsive miRNAs, and target genes from the biological pathway network (* *p* < 0.05): (**a**) Changes of expression in plant samples in response to pathogens visualized using MapMan. (**b**) Changed expression levels of identified unigenes and microRNAs (**c**) under different conditions.

## 4. Discussion

Cadmium has been considered with increasing frequency worldwide as a toxicant, and it is a serious and urgent environmental problem. Meanwhile, Cd also poses a threat to human health as a result of accumulation in the human body via the food chain, leading to chronic toxicity in bones, lungs, kidney and other organs [45,46]. Considering these detriments of Cd, many studies have applied a high-throughput sequencing approach to identify dysregulated genes and miRNAs associated with Cd stress in several species [9,33]. Notably, a number of miRNAs and target genes responding to Cd stress have been identified in some species. However, few studies have analyzed Cd stress-associated mRNAs and miRNAs in *Brassica juncea*. In this report, approximate 16 and eight million raw reads were obtained from transcriptome and sRNA libraries of *Brassica juncea* leaves and roots with or without Cd treatment, respectively.

High-throughput sequencing technologies have been applied to obtain comprehensive sequencing data for transcriptome and non-coding RNA analysis, which may help elucidate the molecular mechanisms at play in associated biological processes and pathways [33,47]. In the current study, 4 transcriptomes and four sRNA libraries were constructed from the leaves and roots of Cd-treated and normal control in *Brassica juncea*, and sequence data were generated by the Illumina platform to analyse the miRNA–unigene regulatory pathway and genes networks in response to Cd stress. Transcriptomic pooling from the leaves and roots of *Brassica juncea* with or without Cd treatments has provided more valuable information to facilitate the screening of Cd-responsive miRNAs and their targets [6]. We found that the differentially expressed unigenes identified from roots and leaves were mainly down-regulated after Cd treatment, which was consistent with previous studies [24]. Importantly, we found that aberrantly expressed unigenes were significantly enriched in a metabolic process that was severely affected by Cd stress. The unigenes affect metabolic processes by interacting with organic compounds when $Cd^{2+}$ enters plant cells or interacts with lipids and proteins through oxidative stress [48]. These findings suggest that the down-regulation of unigenes might play a crucial role in plants' responses to Cd stress.

Moreover, 59 conserved miRNAs and 93 novel miRNAs were successfully screened as associated with Cd stress in *Brassica juncea*. Among them, 57 miRNAs were down-regulated, while 95 were up-regulated, in response to Cd exposure, indicating that miRNAs are differentially regulated under heavy metal stress. Various published reports have identified the many responses of miRNAs to diverse heavy metals, such as Cd, Hg, As and Al in soybean [42], rice [43], *Brassica juncea* [44] and *M. truncatula* [17], which has provided valuable information regarding the regulatory mechanisms of plant miRNAs in response to heavy metal stresses.

In this report, conserved microRNA families, including miR156, miR159, miR166 and miR398, were differentially regulated in response to Cd stress, which agreed with the findings for rice and *Brassica napus* [43,44]. We also observed that some conserved miRNA families, such as miR172, miR161 and miR162, were dysregulated in response to Cd stress, but there was no significantly differential expression after Cd treatment in radishes [24]. Notably, some previously reported miRNA families in responses to heavy metal stress showed temporal organ-specific expression patterns. In a previous report, miR161, miR171, miR398, miR319 and miR385 were differentially expressed in roots and shoots [17]. Similarly, we found that only the miR857 family was expressed in the root, while miR9408 was expressed in the leaf. Therefore, further studies will be needed to explore the molecular mechanisms of those miRNAs in plants' responses to Cd stress.

miRNA is a major regulator of target gene expression at the post-transcriptional level. A great number of dysregulated miRNAs and their target genes exert key roles in response to plant organs under heavy metal stress [24,42,44]. One miRNA named bra-miR172b-3p was found to target 15 transcripts, and also showed significant differential expression between two contrasting Finger millet genotypes while under salinity stress [49], which was in line with our results in this study. Moreover, bra-miR172b-3p has been found to

be down-regulated under Turnip mosaic virus stress in non-heading Chinese cabbage, providing a better way to understand the relationship between plants and virus [50].

Surprisingly, several key enzymes associated with heavy metal ($Cd^{2+}$, $Hg^{2+}$, $Mn^{2+}$, $Pb^{2+}$ or $Na^{+}$) uptake and translocation were found to be target genes for a few conserved miRNAs. bra-miR172b-3p targeted several gene-encoding enzymes related to abiotic stress, such as *DOX1*, *ATCCS*, *ATCAT3* and *ATKT1* (Figure 7a). Functional annotation analysis indicated that these target genes participate in various stress responses, including abiotic stress. It is well known that production of reactive oxygen species in plants is closely related to the phytotoxicity caused by heavy metals, and toxic concentrations of high Cd also induce oxidative stress in plant [51,52].

Although Cd is not a redox-active element, increased reactive oxygen species (ROS) levels may be induced via indirect stress or a maladjustment of the antioxidant system, such as the induction of enzymatic lipid peroxidation and the activation of ROS-producing enzymes [53,54]. Moreover, superoxide dismutase (SOD) and catalase have been widely considered as important defense systems of plants against oxygen free radicals. SOD as an antioxidant enzyme in physiological mechanism and defense system against high concentrations of ROS, and is associated with enhanced tolerance and resistance to oxidative stress [55,56], indicating the critical role of *ATCCS* in abiotic stress regulated by miRNAs. A previous study demonstrated the effects of aluminum on lipid peroxidation and the activities of enzymes associated with the production of activated oxygen stress [6].

Notably, a certain degree of Cd accumulation is involved in the induction of oxidative response in *Holcuslanatus* and *Saccharomyces cerevisiae*, immediately activated antioxidant enzyme to resist the stress response [57,58]. Recently, the key role of SOD in the detoxification process against Cd in *Paxillusinvolutus* has been demonstrated using high-throughput sequencing [59]. Meanwhile, *CCS* delivers heavy metals to SOD in cytosol [60]. Similar to copper chaperone for superoxide dismutase, ATCCS plays an important role in the Cd defense by regulating the levels of SOD in plants. These findings suggest that bra-miR172b-3p could be involved in the Cd defense against oxygen free radicals by directing the regulation of *DOX1*, *ATCCS*, *ATCAT3* and *ATKT1* in *Brassica juncea*. Notably, a recent study has demonstrated the role of miR172 in regulating tobacco tolerance to high salt environment (NaCl), which was consistent with our qRT-PCR results showing that the level of bra-miR172b-3p significantly decreased after NaCl treatment [61]. By targeting complementary sequences, miRNA negatively regulates targeted gene expression, the identified miRNA-mediated gene expression of *DOX1, ATCCS, ATCAT3,* and *ATKT1* could play a vital role in the regulatory networks of Cd detoxification in *Brassica juncea*.

Although RNA-seq generally offers higher detection sensitivity and opens a new avenue in the research on gene fusions, novel alternative transcripts and allele-specific expression, some limitations also exist. The nonconformity between RNA-seq and qRT-PCR verification for bra-miR398-3p suggests a gene-specific biases in high-throughput sequencing, but qRT-PCR should be considered for the final detection of interesting miRNAs or target genes [62]. Meanwhile, only one sample group was detected in the RNA-seq experiment, and three replicate tests were conducted in qRT-PCR due to limited funds. Therefore, this nonconformity may be a false positive because of the individual differences in various samples.

## 5. Conclusions

This study achieved the genome-wide identification of Cd-responsive conserved and novel miRNAs and their target genes in *Brassica juncea* via transcriptomic and sRNA-sequencing technology. In total, 11 miRNAs and 56 transcripts were identified and considered as Cd-related miRNAs and transcripts. Several targets were functionally predicted to encode stress-responsive enzymes or proteins. Specifically, bra-miR172b-3p plays a key role in the response to Cd stress via regulating *ATCCS*. These findings may deepen our understanding of the regulatory relationship between miRNA and transcripts in *Brassica*

*juncea* in response to Cd stress, and provide further insight into the molecular mechanisms of Cd stress in plants.

**Supplementary Materials:** The following are available online at https://www.mdpi.com/article/10.3390/pr9071099/s1, Table S1: Primers for qRT-PCR.

**Author Contributions:** Conceptualization, L.L., D.Z. and M.L.; methodology, H.Y. and L.L.; software and analysis, H.Y. and L.S.; investigation, Y.L., L.S. and X.Y.; data curation, H.Y. and L.L.; writing—original draft preparation, L.L. and H.Y.; writing—review and editing, L.L. and M.Y. funding acquisition, L.L. All authors have read and agreed to the published version of the manuscript.

**Funding:** The Young Scholar Fund of Hunan Provincial Education Department in China (2016601).

**Institutional Review Board Statement:** Not applicable.

**Informed Consent Statement:** Not applicable.

**Data Availability Statement:** The data that support the findings of this study are available at http://www.mirbase.org/, February 2020. These data were derived from the following resources available in the public domain: https://www.ncbi.nlm.nih.gov/, March 2020; https://sourceforge.net/projects/mireap/, March 2020.

**Conflicts of Interest:** The authors declare no conflict of interest.

**Abbreviations**

| | |
|---|---|
| BP | Biological process. |
| CC | Cellular component |
| Cd | Cadmium |
| COG | Cluster of Orthologous Groups |
| DOX1 | Dioxygenases1 |
| EST | Expressed sequence tag |
| FET | Fisher's exact test |
| FLA9 | Fasciclin-Like Abrabinogalactan-protein 9 |
| GDPD | Glycerophosphodiester phosphodiesterase |
| GO | Gene ontology |
| GSS | Genomic survey sequences |
| HMA2 | Heavy metal ATPase 2 |
| HMA3 | Heavy metal ATPase 3 |
| KEGG | Kyoto Encyclopedia of Genes and Genomes database |
| MF | Molecular function |
| MiRNAs | MicroRNAs |
| MS | Modified half-strength |
| Nramp1 | Natural resistance-associated macrophage proteins 1 |
| Nramp3 | Natural resistance-associated macrophage proteins 3 |
| nt | Nucleotides |
| PAP | Purple acid phosphatase |
| qRT-PCR | Quantitative real-time PCR |
| Rfam | RNA family |
| ROS | Reactive oxygen species |
| RPKM | Reads Per Kilo bases per Million reads |
| Rrna | Ribosomal RNAs |
| snoRNAs | Small nuclear RNAs |
| SOD | Superoxide dismutase |
| sRNA | Small endogenous non-coding regulatory RNA |
| TPM | Transcripts per million |
| TRNAs | Transfer RNAs |

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
