# Peer review of "Analysis of Cadmium-Stress-Induced microRNAs and Their Targets Reveals bra-miR172b-3p as a Potential Cd2+-Specific Resistance Factor in Brassica juncea"

_processes, doi:10.3390/pr9071099_

Round 1

Reviewer 1 Report

Overall, the manuscript addresses an environmental issue and a genetical solution way and detailed manner and thus adds up additional information to the field of the biological aspect of heavy metals pollution in the environment.

There are some comments to improve the manuscript, as below:

  • Please give a thorough grammar check to the entire manuscript. There are some mistakes are addressed in the reviewed pdf file.
  • Page 1, line 40, it is better to not use abbreviation at the start of the sentence. Please correct it
  • Page 8, figure 3, please abbreviate the 25 COG categories, by this way the results are presented in a better way.
  • Why didn't you do statistical analysis and comparison of means? For example, in figures 3 and 4 the significance of the difference between COG categories is not clean
  • Try to incorporate some more update references to validate the results.
  • In a separate session, please discuss why did you select Brassica juncea for this research
  • In the abstract, the main quantitative results of the research must be highlighted. Please review the abstract.
  • The introduction is small and poor in works about the works on the Brassica juncea in recent years. Please use some update references
  • What is the novelty of your research? In recent years some works have been done in this field and according to your findings, a novel result should be found in this research. Please explain the novelty of this work at the end of the introduction part.
  • A list of acronyms must be provided
  • Please mention the hypotheses of the study in the last section of the introduction.
  • The writing of the manuscript is poor. The manuscript should be reviewed by a native or an English expert.

Finally, after doing all of these comments and improve the English of the manuscript, it is suggested to consider this manuscript in the final review and publish process.

Author Response

Dear reviewer,

I have made some replies for your some valuable comments. Thank you very much. 

(1) Page 1, line 40, it is better to not use abbreviation at the start of the sentence. Please correct it

Reply: It has been revised.

(2) Page 8, figure 3, please abbreviate the 25 COG categories, by this way the results are presented in a better way.

Reply: The Figure 3 was revised.

(3) Why didn't you do statistical analysis and comparison of means? For example, in figures 3 and 4 the significance of the difference between COG categories is not clean.

Reply: The statistical analysis has done using the P value of the enrichment with Fisher's exact test (as described in method 2.4), not the P value of the t-test. The similar analytical methods and images have been reported in some literature, For example, Wei,et al.,2014; Sun et al.,2015;Yu,et al.,2015,2016; Fu, et al.,2019, and so on.

Wei, W.; Wu, X.G; Liu, Z.J.; Zheng, HJ, and Cheng, Y.X. Insights into Hepatopancreatic Functions for Nutrition Metabolism and Ovarian Development in the Crab Portunus trituberculatus: Gene Discovery in the Comparative Transcriptome of Different Hepatopancreas Stages, PLoS ONE, 2014.

Sun X, Xu L, Wang Y, Yu R, Zhu X, Luo X, Gong Y, Wang R, Limera C, Zhang K, Liu L. Identification of novel and salt-responsive miRNAs to explore miRNA-mediated regulatory network of salt stress response in radish (Raphanus sativus L.).BMC Genomics. 2015 Mar 17;16(1):197. doi: 10.1186/s12864-015-1416-5.

Yu R, Wang Y, Xu L, Zhu X, Zhang W, Wang R, Gong Y, Limera C, Liu L. Transcriptome profiling of root microRNAs reveals novel insights into taproot thickening in radish (Raphanus sativus L.). BMC Plant Biol. 2015 Feb 3;15:30. doi:10.1186/s12870- 015-0427-3.

Yu R, Wang J, Xu L, Wang Y, Wang R, Zhu X, Sun X, Luo X, Xie Y, Everlyne M, Liu L.Transcriptome profiling of taproot reveals complex regulatory networks during taproot thickening in radish (Raphanus sativus L.). Front Plant Sci. 2016 Aug 22;7:1210. doi: 10.3389/fpls.2016.01210. eCollection 2016.

Fu, Y; Mason, A.S; Zhang, Y.F., Lin, B.G.; Xiao, M.L.; Fu, D.H.; Yu, H.S. MicroRNA-mRNA expression profiles and their potential role in cadmium stress response in Brassica napus. BMC Plant Biology 2019,19, 570. https://doi.org/10.1186/ s12870-019-2189-9

(4) Try to incorporate some more update references to validate the results.

Reply:  It has been incorporated some update references.

(5) In a separate session, please discuss why did you select Brassica juncea for this research.

Reply: Brassica juncea L. (AABB, 2n=36) is an amphidiploid species that originated from interspecies crosses between Brassica nigra (AA, 2n=16) and Brassica rapa (BB, 2n=20), which is another important annual or biennial oil crop and leafy vegetable used worldwide [10-11]. Brassica juncea is also a cruciferous plant capable of accumulating significant quantities of metals, including Cd, Zn, Cr, Cu, Au, Pb and Se [12-14]. The plant can accumulate > 400 μg Cd/g dry weight in its leaves over 96 h, and it therefore holds promise for the remediation of Cd-contaminated soils, given its high biomass, moderate metal accumulation capacity, ease of harvesting, and metal tolerance [15].

(6) In the abstract, the main quantitative results of the research must be highlighted. Please review the abstract.

Reply: It has been revised.

(7)The introduction is small and poor in works about the works on the Brassica juncea in recent years. Please use some update references.

Reply: The introduction was revised by using some update references.

(8) What is the novelty of your research? In recent years some works have been done in this field and according to your findings, a novel result should be found in this research. Please explain the novelty of this work at the end of the introduction part.

Reply: Our previous study has investigated the physiological responses of Brassica juncea L. to 30 and 50 mg/kg Cd stress, and demonstrated the activities of catalase enzymes. As the contents of soluble sugar and chlorophyll were reduced, the content of the soluble protein malondialdehyde increased [6]. Moreover, comparative transcriptomic analysis indicated that the downregulation of HMA3 and the upregulation of Nramp3, HMA2 and Nramp1 also play roles in reducing Cd toxicity in the roots of Brassica juncea L. under Cd stress. However, the activity of cadmium in response to miRNAs and their target genes, as well as their corresponding molecular pathways, in Brassica juncea have yet to be fully elucidated. A combined analysis of microRNA and mRNA expression to infer Cd-induced regulation has not been performed for Brassica juncea. In the current study, with the aim of identifying the Cd2+-regulated unigenes and miRNAs in roots and leaves and developing a Cd2+-associated miRNA regulatory network, we constructed mRNA and sRNA libraries of Cd2+-treated and Cd2+-free Brassica juncea, which were then sequenced via an Illumina Hiseq2500 system. These results could provide a new perspective for studying the regulation of miRNAs in root and leaf under heavy metal stress, and facilitate genetic breeding of cadmium tolerance in Brassica juncea.

(9) A list of acronyms must be provided.

Reply: The list of acronyms were provided.

(10) Please mention the hypotheses of the study in the last section of the introduction.

Reply: The introduction has been improved.

(11) The writing of the manuscript is poor. The manuscript should be reviewed by a native or an English expert. Finally, after doing all of these comments and improve the English of the manuscript, it is suggested to consider this manuscript in the final review and publish process.

Reply: The manuscript was edited by MDPI language service.

Reviewer 2 Report

I have reviewed the manuscript and please find suggestion on the manuscript of Liu et al. entitled " Analysis for cadmium stress-induced microRNAs and their targets reveals bra-miR172b-3p as a potential Cd2+-specific resistant factor in Brassica juncea

The present study evaluates the effect of the cadmium stress on the expression of miRNA in Brassica juncea. The research conducted in this study was well supported by the experimental and statistical findings. The manuscript has some inconsistencies in terms of formatting and English. The scientific content of the manuscript is good but needs to be presented in a better way and should be addressed before publication in journal Processes. It needs careful reading before submission. My minor concerns are listed in the manuscript itself. Please look at the comments throughout the manuscript and needs to be addressed before acceptance in the manuscript.

My major concerns are as follows:

  1. The general technique used for assessing the expression of miRNA is stem-loop PCR, which involve universal reverse primer and specific miRNA primer. I would like to confirm whether the authors used same technique or if modified the protocol. The authors should cite the reference of the method followed.
  2. Authors did not mention about Cd2+ stress-responsive transcriptome and their expression. For eg., The expression of At3g01420 (dioxygenases, DOX1) was assessed by the real-time PCR as a target of one of the identified miRNA. But authors have conducted transcriptome analysis too, what is the expression pattern of these target genes in NGS expression analysis?
  3. There are few spell checks throughout the manuscript and need proper formatting.

Author Response

Dear reviewer,

I have made some replies for your some valuable comments. Thank you very much. 

My major concerns are as follows:

  1. The general technique used for assessing the expression of miRNA is stem-loop PCR, which involve universal reverse primer and specific miRNA primer. I would like to confirm whether the authors used same technique or if modified the protocol. The authors should cite the reference of the method followed.

Reply: Some reference was cited.

  1. Authors did not mention about Cd2+stress-responsive transcriptome and their expression. For eg., The expression of At3g01420 (dioxygenases, DOX1) was assessed by the real-time PCR as a target of one of the identified miRNA. But authors have conducted transcriptome analysis too, what is the expression pattern of these target genes in NGS expression analysis?

Reply: Reply: In the Figure 7b, the first column ("black" for RNA-Seq ) in every target genes (ATCCS, AT3G04320, FLA9, DOX1, ATCAT3 and ATKT1) are about Cd2+ stress-responsive transcriptome ( RNA-seq) and their expression. Based on the results of miRNA-seq and RNA-seq, bra-mir172b-3p was down-regulated, and its target genes DOX1, ATCCS, ATCAT3 were significantly up-regulated expression in Brassica juncea under Cd stress, which was in line with our results by RT-PCR analysis (Fig. 7b-c).Then, the expression levels of six selected target genes—ATCCS, AT3G04320, FLA9, DOX1, ATCAT3 and ATKT1—and three miRNAs, including bra-miR172b-3p, mireap-2590 and bra-miR398-3p, in response to Cd2+, Hg2+, Mn2+, Pb2+ and Na+ stress and Mucor infection were validated via qRT-PCR analysis. The result 3.6 and discussion were supplemented.

  1. There are few spell checks throughout the manuscript and need proper formatting.

Reply: The manuscript was edited by MDPI language service.

Round 2

Reviewer 1 Report

The authors improved the paper. But I have some comments. Pls see it in the attachment.

Author Response

Dear reviewer,

I have made some revise in the updated manuscripts. Thank you very much.